# Deciphering Molecular Mechanisms Involved in the Modulation of Human Aquaporins’ Water Permeability by Zinc Cations: A Molecular Dynamics Approach

**DOI:** 10.3390/ijms25042267

**Published:** 2024-02-14

**Authors:** Robin Mom, Stéphane Réty, Vincent Mocquet, Daniel Auguin

**Affiliations:** 1Laboratoire de Biologie et Modélisation de la Cellule, ENS de Lyon, Université Claude Bernard, CNRS UMR 5239, INSERM U1293, 46 Allée d’Italie Site Jacques Monod, F-69007 Lyon, France; stephane.rety@ens-lyon.fr (S.R.); vincent.mocquet@ens-lyon.fr (V.M.); 2Research Group on Vestibular Pathophysiology, CNRS, Unit GDR2074, F-13331 Marseille, France; 3Laboratoire de Physiologie, Ecologie et Environnement (P2E), UPRES EA 1207/USC INRAE-1328, UFR Sciences et Techniques, Université d’Orléans, F-45067 Orléans, France

**Keywords:** aquaporin, zinc, AQP0, AQP2, AQP4, AQP5, molecular dynamics

## Abstract

Aquaporins (AQPs) constitute a wide family of water channels implicated in all kind of physiological processes. Zinc is the second most abundant trace element in the human body and a few studies have highlighted regulation of AQP0 and AQP4 by zinc. In the present work, we addressed the putative regulation of AQPs by zinc cations in silico through molecular dynamics simulations of human AQP0, AQP2, AQP4, and AQP5. Our results align with other scales of study and several in vitro techniques, hence strengthening the reliability of this regulation by zinc. We also described two distinct putative molecular mechanisms associated with the increase or decrease in AQPs’ water permeability after zinc binding. In association with other studies, our work will help deciphering the interaction networks existing between zinc and channel proteins.

## 1. Introduction

Zinc is the second most abundant trace element in the human body and is known for its multiple functions in the body’s physiology [1]. Because zinc cations are associated with a very wide range of proteins involved in many different metabolic pathways and structural roles, its deficiency is associated with broad symptoms including growth limitations, persistent diarrhea, or altered resistance to infections [2,3]. Many biochemical functions have been discovered for zinc cations and include three main roles: a structural role [4], a catalytic role [5], and a role in the maintenance of plasma membranes’ functions [6]. Zinc is known to interact with protein motifs called zinc fingers found in many transcription factors. These motifs are characterized by cysteine and histidine residues positioned close to each other and forming the zinc fixation site [7].

Aquaporins are transmembrane channel proteins dedicated to the facilitation of water and the passage of other small polar solutes across biological membranes [8,9,10,11]. They have been implicated in numerous pathologies and multiple attempts have been made to develop AQPs as new therapeutic targets [12,13,14]. AQPs are naturally found in tetrameric assemblies with each of the four sub-units being functional water channels (Figure 1A). The structural hourglass shape of the AQP protomer allows for a very constrained and specific channeling of water molecules, which are forced to form a single file continuum within the conducting pore (Figure 1A) [8,10]. In the narrow water pore, the nature and location of residue side-chains contribute to local constrictions such as the aromatic/arginine (ar/R) constriction, which is the narrowest part of the pore in most AQPs. The center of the channel is located at the meeting point of the two asparagine-proline-alanine (NPA) motifs. In AQP0, two additional constrictions are caused by tyrosine 23 and tyrosine 149 side-chains (Figure 1A), making AQP0 the human AQP with the lowest water permeability [15].

Several studies indicate that a regulation of human AQP functions by zinc cations could exist [16,17,18]. AQP0 is the main AQP found in the eye [19]. Zinc is also known to be particularly high in human ocular tissues, where it plays a wide range of roles and is mandatory for the maintenance of normal ocular functions [20]. Moreover, it has been shown that zinc cations could induce significantly higher water permeability of AQP0 expressed in *Xenopus laevis* oocytes [16]. AQP5 is mainly expressed in secretory glands, where it plays a central role in fluid secretion [19,21]. Interestingly, another study has demonstrated that zinc supplementation could induce an increase in these glands’ secretory function [18]. AQP4 is the most abundant AQP of the brain [22]. In the central nervous system (CNS), zinc has been implicated in controlling synaptic excitability, neuronal death, or the development of amyloid plaques in Alzheimer’s disease [23]. Contrarily to AQP0, AQP4 has been shown to be transiently inhibited by zinc cations in reconstituted proteoliposomes [17]. Finally, AQP2 is a ubiquitous AQP well known for its functions in renal water re-absorption [19]. No direct regulation of AQP2 by zinc has been yet described; however, zinc deficiency is also associated with kidney diseases [24].

In the present study, we aimed at better understanding the nature of zinc cations and AQP interactions. In order to do so, we investigated the impact of zinc fixation on AQP0, AQP2, AQP4, and AQP5 at the molecular level through molecular dynamics simulations. Our results for AQP0 and AQP4 align with other approaches and strengthen the relevance of zinc as a modulator of human AQP functions. Moreover, we found evidence in favor of direct regulation of two other AQPs by zinc cations: AQP5 and AQP2. Hence, this regulation seems to be a conserved feature among AQPs and could help to explain the very wide range of zinc actions in human physiology. From this preliminary set of results, it appears that zinc can act upon AQPs in two different ways involving two distinct molecular mechanisms and fixation sites. Finally, we discuss the reliability of our results and their implications for AQP regulation and human physiology.

## 2. Results

All of our predicted zinc binding sites were defined by the presence of a solvent accessible cysteine. For both AQP0 and AQP5, these putative zinc binding sites are located at the interfaces between two sub-units and we hypothesized that they would be accessible to zinc cations from the extra-cellular compartment through the central pore of the tetramer (Figure 1B,C). For both AQP4 and AQP2, on the other hand, our predicted zinc binding sites are localized on the intra-cellular surface of the tetramer and should be directly accessible to zinc cations from the cytoplasm (Figure 1D–G).

### 2.1. Zinc Increases AQP0 and AQP5 Water Permeability

AQP0 and AQP5 both displayed similar responses to zinc fixation (Figure 2). In both cases, a significant enhancement of their water permeability was measured (Figure 2A). This phenotype correlated perfectly with a significant reduction in the highest free-energy barrier on free-energy profiles of water molecules inside the conducting pores (Figure 2B). This barrier was coherently situated at the location of a known pore constriction: at the extra-cellular tyrosine 23 constriction for AQP0 or at the extra-cellular ar/R constriction for AQP5. This lowering of the free-energy barrier also correlated with a locally significant increase in the pore radius (Figure 2C).

### 2.2. Molecular Mechanism of AQP0 and AQP5 Water Permeability Enhancement by Zinc

AQP0 was initially in a non-functional conformation (and stayed that way in the control condition; see Figure 2A). When zinc cations were added, however, one of the sub-units became fully functional. This is clearly illustrated by the cumulative number of water molecules crossing the conducting pore (Figure 3A). One sub-unit, chain A, became functional after approximately 20 nanoseconds of simulation, while the three other chains were kept closed. The putative zinc fixation motif for AQP0 was located at the interface between two sub-units (or chains). The position of a zinc cation at this location induced a slight change in helices’ orientations (Figure 3B). This can be visualized on the root mean square fluctuation (RMSF) profiles for the helices in direct interaction with zinc cations (Figure 3C). Among these helices, helix 1 and helix 3 bear the two tyrosines responsible for low AQP0 permeability. The changes in orientation of the AQP helices hence induce a small change in tyrosine 23 side-chain position inside the lumen of the pore, which allows for a widening of the constriction and an increase in the water fluxes. The RMSF of tyrosine 23 alpha carbon indicates that a very small change in alpha carbon position (~0.05 angströms) (Figure 3C) can be associated with side-chain repositioning, leading to an increase in pore diameter of ~0.5 angströms (Figure 2). Tyrosine 23 was associated with the low water permeability of AQP0 (see Figure 2) and is located on helix 3. Hence, we compared the interface containing chain A helix 3 (which corresponded to an open channel) to the interface containing chain D helix 3 (corresponding to a closed channel) (Figure 3D). For both interfaces, we looked at the smallest distance between residues glutamine 140 (Q140), cysteine 144 (C144), threonine 54 (T54), glutamine 57 (Q57), and zinc (Figure 3E). For the interface corresponding to an open channel, the zinc cation was positioned sufficiently close to C144, Q140, and T54 (<2.5 angströms) for the formation of a salt bridge network that linked chain C and chain A together. On the other interface however, the zinc cation was stabilized by residues belonging to the same chain (C144, serine 167, or threonine 148; Figure 3D,E). Indeed, both residues from the adjacent sub-unit (T54 and Q57) were kept at a distance of about 4 angströms from the zinc cation, hence preventing the formation of a link between chain A and chain D.

The enhancement of AQP5 water permeability by zinc correlated with a lower free-energy barrier (Figure 2B) and the increase in the pore radius (Figure 2C) at the ar/R constriction. This is further confirmed by the very significant increase in the smallest distance between R188 and H173 of this same constriction (Figure 4A). Through similar molecular dynamics studies, it was shown that extra-cellular loop C conformational changes were significantly associated with the channel water permeability of human AQP2 [25,26]. The underlying hypothesis was that residues of loop C directly interacting with the arginine of the ar/R constriction through hydrogen bonds could finely tune the position of such arginine side-chain inside the pore lumen. Hence, we tested the relevance of this first hypothesis by following the number of hydrogen bonds formed and the minimal distance between the AQP5 corresponding residues: between ar/R arginine 188 and loop C residues alanine 118 and asparagine 120 (Figure 4A–C). A small but statistically significant decrease in hydrogen bond formation was observed in “zinc” condition compared with “control” (Figure 4C). This correlated with an equally small but significant increase in R188-A118 and R188-N120 minimal distances (Figure 4A). Moreover, root mean square fluctuations of the small helix HE backbone were in good accordance with AQP0 results as the ar/R arginine 188 alpha carbon was also displaced by ~0.05 angströms (Figure 4C). These results confirm the implication of loop C in the regulation of human AQPs’ water permeability through its role in the stabilization of the ar/R arginine side-chain within the pore lumen. However, more research is needed to better understand the exact significance of this loop in AQP water permeability tuning.

To better understand how AQP5 is regulated by zinc, we looked at each sub-unit separately and compared them to the corresponding sub-unit in the “control” (i.e., without zinc) condition (Figure 5). We observed phenotypic heterogeneity between sub-units, with chain A and chain D displaying the most significant increase in water permeability (Figure 5A). For each sub-unit, there are two adjacent sub-units. Zinc cations are located in these interfaces and are bound by residues from two adjacent sub-units (Figure 5B). In a similar way as for AQP0, two adjacent sub-units can be linked through their interaction with the zinc cation. We can see that the permeability of a sub-unit correlates with the formation of such a zinc-mediated link with the adjacent sub-units (Figure 5C). Indeed, for both chain A and chain D, which correspond to the most significant increases in water permeability, the two interfaces are in a “cross-linked by zinc” state with the adjacent sub-unit. For example, chain A is linked to the adjacent sub-unit at interface A through a salt bridge network involving its cysteine 145 and its serine 164, and glutamine 58 of the adjacent sub-unit (all forming a salt bridge with zinc cations, as indicated by the smallest distance with zinc of ~2 angströms; Figure 5C). For interface B, chain A is linked to the adjacent sub-unit through a salt bridge network involving its threonine 55, and cysteine 145 and serine 168 of the adjacent sub-unit. For chain B and chain C, however, only one of the two interfaces is in a “cross-linked by zinc” state and both channels display a significantly smaller increase in water permeability.

### 2.3. Zinc Decreases AQP4 and AQP2 Water Permeability

In the opposite manner as for AQP0 and AQP5, zinc cations induced a significant decrease in AQP4 and AQP2 water permeability (Figure 6A). However, this change in water permeability is similarly integrated by the ar/R constriction as indicated by free-energy profiles and pore radius (Figure 6B,C). For AQP4, three putative zinc binding sites were tested but only one, including cysteine 253, was associated with a significant difference from the control condition (Figure 6A). For AQP2, only one putative binding site could be found and was characterized by the presence of two cysteines (Figure 1). Because these two cysteines are positioned very close to each other, even though the cytoplasm is usually considered to be too reductive to allow disulfide bridge formation, we still evaluated the effect of such a link between intra-cellular cysteine 75 and cysteine 79 on water permeability (Figure 6A). It appeared that this disulfide bridge had no significant impact on the water fluxes. On the contrary, the presence of zinc cations correlated with significantly lower water permeability (Figure 6A).

### 2.4. Molecular Mechanism of AQP4 and AQP2 Water Permeability Reduction by Zinc

Thereafter, we focused on this condition where zinc cations were positioned at the putative binding site associated with cysteine 253. Even though the overall impact of zinc cations on AQP4 was associated with a decrease in water permeability of the tetramer, we also observed high phenotypic heterogeneity between sub-units (Figure 7A). In a similar way as for AQP0 and AQP5, this putative zinc binding site of AQP4 is located at the interface between two sub-units (Figure 7B). Again, the water permeability of each sub-unit correlated with the number of interfaces with adjacent sub-units in a “cross-linked by zinc” state. Indeed, chain A was associated with a significant increase in water permeability and corresponded to the only sub-unit with its two interfaces in a “cross-linked by zinc” state. On the contrary, chain B was associated with a significant decrease in water permeability and had its two interfaces in a “separated” state. Finally, chain C and chain D both had only one of the two interfaces in a “cross-linked by zinc” state and displayed intermediate phenotypes (Figure 7).

To better understand the molecular mechanisms involved, we then continued our analysis of three sub-units associated with different functional phenotypes: chain A as a representative of high water permeability channel; chain B as a representative of low water permeability channel; and chain D as a representative of an intermediate phenotype (Figure 8). The root mean square deviation (RMSD) of these three sub-unit backbones indicates that chain A, which is linked to the two adjacent sub-units through a salt bridge network with zinc (Figure 7), is more stable than in the “control” condition (Figure 8A). Chain B and chain D, however, have similar deviations from the initial conformation as in the “control” condition. Consistent with AQP0 and AQP5 results, in chain A only, the alpha carbon of the arginine 216 of the ar/R constriction was moved ~0.02 angströms away from its original position (Figure 8B). Hence, from analysis of both AQP0, AQP5, and chain A of AQP4, we can conclude that the increase in water permeability by zinc is mediated through the stabilization of the AQP fold through the formation of salt bridge networks linking each sub-unit with its two adjacent sub-units. As a result, the small helix HE backbone is slightly displaced by ~0.05 angströms at the location of the arginine of the ar/R constriction inducing a repositioning of its side-chain inside the lumen of the pore of ~0.5 angströms. Because this arginine plays such a significant role in tuning the channel water permeability (this constriction corresponds to both the narrowest part of the conducting pore and to the highest free-energy barrier of pore water free-energy profiles), such a small change is enough to significantly increase the channel water permeability.

In the sub-unit displaying a significant decrease in water permeability (chain B), however, no link between the adjacent sub-units is established and there is no change in stability or in small helix HE backbone position at the ar/R constriction compared to the “control” condition (Figure 8). We then hypothesized an action on the water permeability through indirect long-range electrostatic interactions with the arginine of the ar/R constriction. Indeed, the arginine 216 of the ar/R constriction has its side-chain protruding within the pore lumen in regards to an aromatic residue. Hence, the positive charge of the guanidinium group cannot be stabilized through the establishment of a salt bridge with other charged residues. This particularity makes the side-chain of R216 especially susceptible to changes in the electrostatic potential of the protein. This is illustrated by the correlation between the vector joining the R216 alpha carbon and the CZ carbon of its guanidinium group and the dipole moment of the sub-unit (Figure 8C). We can see that, for all three sub-units, the correlation is very high even though chain A displays a slightly higher correlation that could be explained by its higher stability. When we looked at the correlation of the sub-unit dipole moment between conditions “control” and “with zinc at the C253 predicted binding site”, very significant differences appeared between chain A and the two other chains (Figure 8C). Even though in all cases the correlation is maintained at very high levels (Pearson coefficient > 0.9), this small shift in correlation indicates a reorientation of the sub-unit dipole moment and a subsequent repositioning of R216 side-chain into the water pore lumen (Figure 8C).

Finally, for AQP2, the position of the zinc putative binding site is not localized at the interface with adjacent sub-units as it was the case for AQP0, AQP5, and AQP4. This is in good accordance with the previously exposed molecular mechanisms associated with water permeability modulation by zinc cations. As expected, for AQP2, the overall stability of the sub-units was not increased compared to the “control” condition. This observation strengthens the link previously highlighted between the zinc-mediated junction to the adjacent sub-units and the stability of the sub-unit (Figure 9A). Coherently, the alpha carbon of ar/R arginine 216 did not fluctuate more or less than in the “control” condition, except for AQP2 chain C, which displays a 0.02-angström drift comparable to AQP4 chain A (Figure 9B). This chain C was, thereafter, the only one that did not display a significantly smaller water permeability than in the “control” condition (Figure 9C). Finally, the correlation coefficients for the dipole moment of sub-units in the “control” condition compared to “zinc” condition correlate again with the water permeability phenotypes (Figure 9C). Indeed, the most significantly divergent dipole moments are associated with the most significant decrease in water permeability (Figure 9C).

## 3. Discussion

In the present study, four human AQPs were investigated for their putative regulation by zinc cations. In all of the conditions studied, zinc cations were maintained in their predicted binding sites during the whole 200 nanoseconds simulated (Appendix A). For AQP0 and AQP5, the addition of zinc cations was associated with significantly higher water permeabilities (Figure 2). On the other hand, for AQP4 and AQP2, zinc cations triggered a significant decrease in water permeabilities (Figure 6). Both of these results align with heterologous expression studies of AQP0 [16] (expressed and characterized in Xenopus laevis oocytes) and AQP4 [17] (reconstituted into proteoliposomes) regulation by zinc. Based on our results, the increase in AQP water permeability is associated with an increase in the overall AQP fold stability (Figure 8), which correlates significantly with the establishment of a network of zinc-mediated salt bridges that link each sub-unit (Figure 3, Figure 5 and Figure 7). A decrease in AQP water permeability mediated by zinc, on the other hand, does not seem to involve such conformational changes (Figure 7, Figure 8 and Figure 9). In both cases, however, zinc binding is eventually functionally integrated through the repositioning of the side-chain of the arginine of the ar/R constriction inside the pore lumen (Figure 2, Figure 4 and Figure 6). This allosteric effect could either be mediated by conformational changes in extra-cellular loop C (Figure 4) or by direct long-range electrostatic interactions (Figure 8 and Figure 9) for water flux increases and decreases, respectively. The implication of loop C in ar/R arginine side-chain position tuning is consistent with two other similar in silico studies [25,26]. Direct sensing of the ionic environment by the arginine of the ar/R constriction of AQPs has already been observed through other molecular dynamics studies for human AQP1 [27], AQP4 [27], or other plant and prokaryotic AQPs [28]. For all the AQPs studied in the present work, a very small change in the pore’s narrowest constriction (Y23 for AQP0 or ar/R constriction for AQP2, AQP4, and AQP5) diameter of about 0.5 angströms was enough to trigger a significant reduction or increase in water permeability (Figure 2 and Figure 6). This repositioning of the tyrosine or arginine side-chain was associated with a 0.05-angström displacement of the alpha carbon of the backbone in AQP0 and AQP5 (Figure 3 and Figure 4). Similarly small positional changes in the side-chain of ar/R constriction residues have already been associated with water flux modulation for AQPs through molecular dynamics studies [27,28,29]. Other divalent cations are also known for their AQP regulatory potential. The main one is mercury and is still used as an AQP non-specific inhibitor, even though its non-specificity makes it very toxic for living organisms [30]. The exact molecular mechanism underlying mercury regulation of AQPs is still under investigation; however, we already know that this regulation takes place through the binding of mercury to a cysteine located in the ar/R constriction [31,32,33]. According to molecular dynamics and nuclear magnetic resonance studies, the fixation of mercury cations directly induces a repositioning of ar/R constriction residue side-chains within the pore lumen [32,33]. Coherently, in our study, the regulation of AQP5, AQP4, and AQP2 by zinc is also integrated by the arginine of the ar/R constriction. Hence, zinc cations seem to differ from mercury cations because of their allosteric effect. However, the molecular mechanisms modulated by both cations are tied to the position of the ar/R constriction residues.

Moreover, the molecular mechanism associated with an increase in AQP water permeability was characterized by sub-unit cooperativity (Figure 3, Figure 5 and Figure 7). Likewise, in a study where AQP0 was expressed in Xenopus laevis oocytes, the authors concluded that a zinc-mediated increase in water fluxes required positive cooperativity between sub-units associated with a Hill coefficient of 4 [16]. This also strengthens the relevance of putative zinc binding sites localized at the interface between sub-units as the ones we described for AQP0, AQP5, and AQP4 (Figure 1, Figure 3, Figure 5 and Figure 7). However, in the same study, the authors demonstrated through mutants that histidine 40 replacement by a cysteine and histidine 122 replacement by a glutamine-abolished zinc regulation. They hypothesized that these two histidines could be part of the zinc binding site associated with this regulation [16]. Through our molecular dynamics approach, however, we concluded that both H40 and H122 were not part of the zinc binding site. Nevertheless, these two histidines could also be indirectly linked to zinc modulation of AQP0. Histidine 40 is localized on the extra-cellular surface of the tetramer, bordering the entrance of the central canal (formed by the tetrameric assembly). The putative zinc binding site that we proposed for AQP0 would be accessible to zinc cations from the extra-cellular compartment through the central pore. Hence, the nature of the residues bordering this central pore could also have a very significant impact on the ability of zinc cations to reach out to their binding site. For instance, the electronegativity of cysteines could trap the zinc cations at the entrance of the central pore, therefore preventing its entry within the tetramer. Histidine 122 is also located on the extra-cellular surface of the tetramer of AQP0 and is part of loop C. This loop C has been hypothesized as also playing a role in the regulation of human AQPs [25,26], which could explain why this mutation could interfere with zinc regulation. Regarding AQP4 regulation by zinc, it has been shown through reconstitution of AQP4 tetramers into proteoliposomes that zinc could impair the protein water flux in a rapid and transient manner [17]. Additionally, the authors found that this regulation was associated with a Hill coefficient of 2.44 ± 1, also suggesting cooperative binding, but binding that was less cooperative than for AQP0–zinc interaction (with a Hill coefficient of 4). Indeed, as we have shown that zinc binding at the interface between monomers has a significant effect on the overall stability of the AQP fold (Figure 8), each binding event could modify the stability and hence the other interfaces of the two sub-units involved in its stabilization. For AQP4, however, the mechanism is completely different and while no interface deformation is involved, the modification of one sub-unit dipole moment could also affect the adjacent sub-unit dipole moment. Indeed, each of these dipoles is orientated toward one of the adjacent monomers. This association between two sub-units through their dipole moment is well illustrated by our result for AQP2, where two monomers have very significantly reduced water permeability while the two others are barely impacted (Figure 9C). Finally, the author suggested that AQP4 regulation by zinc would be mediated by intra-cellular cysteine 178 [17]. In our study, we found no significant regulation associated with the cysteine 178 binding site (Figure 6). However, Yukutake et al. suggested this role for cysteine 178 based on the results they obtained for tetramers with cysteine 178 mutated to serine [17]. First of all, they did not test the effect of this type of mutation for cysteine 253. Finally, while serine and cysteine are fairly similar, serine still has a significantly more polar side-chain than cysteine [34]. In our simulations, we observed that while cysteine 178 was not involved in zinc binding, its neighboring aspartate 179 was systematically linked to zinc cations (Figure 7). Cysteine 178 is located at the junction between helix 4 and the flexible intra-cellular loop D. We could also hypothesize that the more polar side-chain of serine could induce the depletion of one helix turn or any other small local conformational re-arrangement, which could impair the ability of aspartate 179 to stabilize zinc cations with cysteine 253.

Finally, AQPs have been clearly identified as therapeutic targets for cancer treatments [35,36]. Interestingly, zinc cations have also been associated with cancer prevention [37] even though the underlying molecular mechanisms are still under investigation. Direct interaction of zinc with AQPs could also account for its anti-proliferation and anti-migration effects on cancer cells [38]. Inhibition of AQP water permeability in such cells could induce a cell volume decrease associated with apoptosis [39]. Interestingly, AQP4 has been demonstrated to be essential for a regulatory volume decrease in astrocytes [40], while many other studies have linked AQPs to cell migration ability [41]. As an illustration of this hypothesis, AQP4 is found in prostate cancers [42], for which zinc has also been identified as an anti-tumor agent [38].

To conclude, our results strongly align with other approaches and scales of study for AQP0 and AQP4 regulation by zinc cations and highlight two completely distinct molecular mechanisms that nonetheless result in the same fine tuning of ar/R constriction pore-lining arginine side-chain. These convergences strengthen the reliability of a physiological regulation of AQP water permeability by zinc and provide additional insights into AQPs’ regulatory molecular mechanisms. Moreover, by confirming zinc as a relevant AQP regulator, these results also provide new exciting hypotheses for cancer treatments using zinc or other cations.

## 4. Materials and Methods

### 4.1. Molecular Dynamics Simulations

All simulations were performed with GROMACS (v.2018.3) [43] in a CHARMM36m force field [44]. The systems were built with a CHARMM-GUI interface [45,46]. A first minimization step was followed by 6 equilibration steps, during which restraints applied to the protein backbone, side-chains, and lipids were progressively removed before the production phase was performed without restraint. Pressure and temperature were kept constant at 1 bar and 310.15 Kelvin, respectively, using the Berendsen method during equilibration and Parrinello–Rahman and Nosé–Hoover methods during production. The Lennard-Jones interaction threshold was set at 12 angströms (Å) and the long-range electrostatic interactions were calculated through the particle mesh Ewald method [47].

For AQP2, AQP4, and AQP5, the initial conformation was retrieved from RCSB Protein Data Bank, i.e.,pdb 4nef [48], pdb 3gd8 [49], and pdb 3d9s [50], respectively. For AQP0, since no experimental structure from homo sapiens was available, a homology model was retrieved from the alphafold protein structure database [51]. N-terminal and C-terminal extremities of AQPs are long and unstructured and naturally folded by interactions with other proteins such as calmodulins [52,53]. Moreover, they can interfere with water fluxes. Hence, to avoid interference with the effect of zinc, each AQP sub-unit was truncated at the same position, i.e., position 6 or 7 in the N-terminal region (before a serine), and after the proline terminating the last helix of the AQP fold in the C-terminal region. The extremities were capped to avoid artificial charges. The tetrameric assemblies were then inserted into a phosphatidylcholine (POPC) bilayer, solvated with transferable intermolecular potential 3 (TIP3) water [54] molecules and 150 mM of KCl. Based on the well-documented involvement of cysteine in zinc binding domains [7], we arbitrarily defined a putative zinc binding domain for each solvent accessible cysteine of AQP0, AQP2, AQP4, and AQP5. One zinc cation was manually positioned in the vicinity of cysteine 144 or cysteine 145 for AQP0 and AQP5, respectively. For AQP2, the cation was positioned close to two intra-cellular cysteines: cysteine 75 and cysteine 79. Since these two cysteines are located very close to each other, an additional system was built without zinc and with disulfide bridges formed between the two cysteines. For AQP4, since three separated cysteines could interact with zinc, one system was built for each putative site. The zinc cation was hence positioned close to three different cysteines depending on the system: cysteine 87, cysteine 178, or cysteine 253. For each AQP, a control condition corresponded to the same protocol without the addition of the four zinc cations. In summary, the following conditions were studied: AQP0_control (AQP0 tetramer without zinc); AQP0_Zn (AQP0 tetramer with four zinc cations docked); AQP5_control (AQP5 tetramer without zinc); AQP5_Zn (AQP5 tetramer with four zinc cations docked); AQP4_control (AQP4 tetramer without zinc); AQP4_Zn_CYS87 (AQP4 tetramer with four zinc cations docked at the vicinity of cysteine 87); AQP4_Zn_CYS178 (AQP4 tetramer with four zinc cations docked at the vicinity of cysteine 178); AQP4_Zn_CYS253 (AQP4 tetramer with four zinc cations docked at the vicinity of cysteine 253); AQP2_control (AQP2 tetramer without zinc); AQP2_SS (AQP2 tetramer with intra-cellular disulfide bridges formed between cysteine 75 and cysteine 79); and AQP2_Zn (AQP2 tetramer with four zinc cations docked).

All the systems were then simulated for 200 nanoseconds.

### 4.2. Analysis

#### 4.2.1. Water Permeability

To monitor water molecules’ displacement along the trajectories, the MDAnalysis library was used [55,56]. Water counts were derived from these water coordinates. One water permeation event corresponds to one water molecule crossing the whole transmembrane section of the conducting pore, i.e., a cylinder having a radius of 15 angströms and a length of 30 angströms centered on the center of geometry of the alpha carbons of the two asparagines of the NPA motifs (which meet at the center of the channel).

#### 4.2.2. Free-Energy Profiles

Water free-energy profiles were extrapolated from the logarithm function of the water counts inside the pore with the *z*-axis as a reaction coordinate [29,57]. The pore is divided along the reaction coordinate (*z* axis) in slices of 0.5 Angströms (Å). The average density of water molecules in each slice was then computed over the 200-nanosecond portions of simulation, and the Gibbs free energy *G*(*z*) was obtained as follows:$$G\left(z\right)=-KTln\frac{\rho (z)}{\rho_{bulk}}$$
where *K*, *ρ_bulk_*, and *T* represent the Boltzmann constant, the bulk density, and the absolute temperature, respectively.

#### 4.2.3. Other Properties

Distances, root mean square deviations, root mean square fluctuations, and dipole moments were computed with GROMACS tools (version 2020.6). Pearson correlation coefficients between dipole moments and/or the vector joining ar/R constriction arginine alpha carbon and guanidinium CZ carbon were obtained as follows: the two vectors of interest were normalized to the origin, and the cosine of the angle formed between the two vectors was used as correlation coefficient.

Pore profiles were computed with HOLE software (release 2.2.005) [58]. All the schematic representations were made with PyMOL software (version 2.5.0) [59].

#### 4.2.4. Statistical Analysis

All statistical analysis were performed using the R programming language. Before any statistical test was performed, normality and homoscedasticity of the variables were controlled to choose between parametric or non-parametric tests. Thereafter, when two conditions were compared, either Student’s T test or the Mann–Whitney test was used, and when more than two conditions were compared, either Tukey’s post hoc test after one-way analysis of variance or Bonferroni post hoc correction after the Wilcoxon test was used.

For each condition, the 200-nanosecond-long whole trajectory was divided into 10-nanosecond sub-trajectories. The four sub-units were taken into account, hence yielding 80 repetitions per condition.

## Figures and Tables

**Figure 1 ijms-25-02267-f001:**
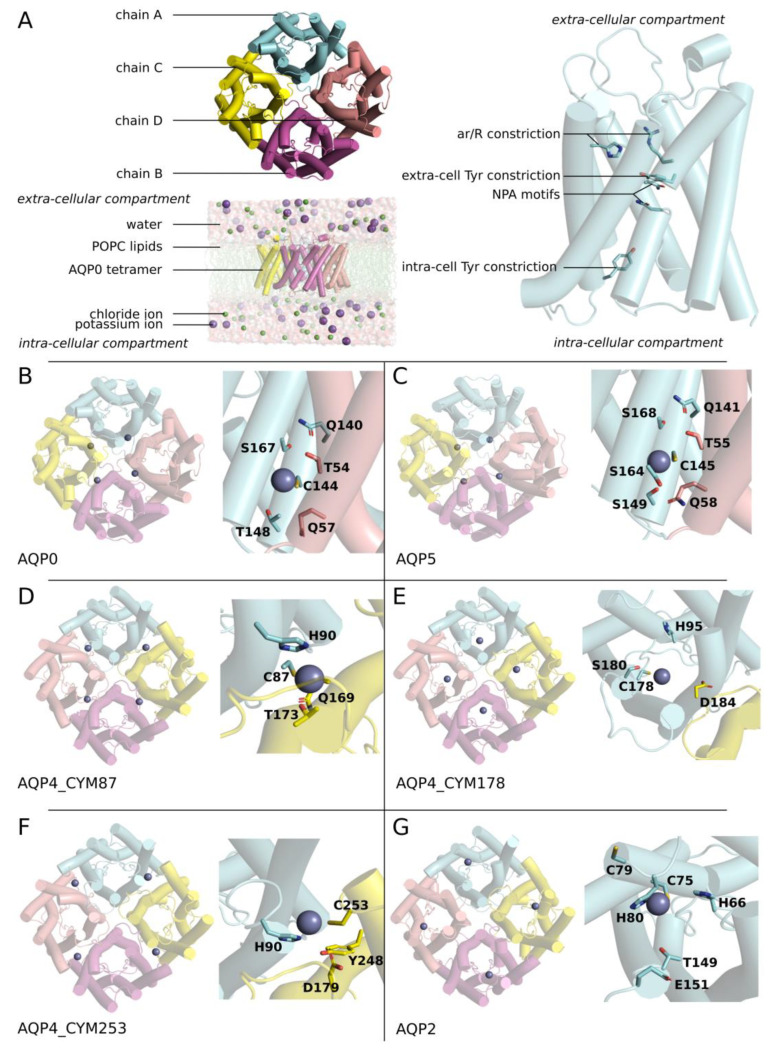
Building of the atomic systems simulated. (**A**) Schematic representation of the atomic systems simulated. AQPs are naturally found in tetrameric assemblies (**upper left** corner). The tetramer of each simulated AQP was inserted into phosphatidylcholine (POPC) bilayer and solvated with transferable intermolecular potential 3 (TIP3) water and 150 mM KCl ions (**lower left** corner). Each sub-unit constitutes a functional water channel with conserved constrictions: the ar/R constriction, the region of NPA motifs, which meet at the center of the pore, and, for AQP0 only, two additional constrictions are induced by a tyrosine protruding within the pore (**right** side). (**B**,**C**) Schematic representation of AQP0 and AQP5 tetramers as seen from the extra-cellular compartment. (**D**–**G**) Schematic representation of AQP4 and AQP2 tetramers as seen from the intra-cellular compartment. For AQP4, three different conditions were simulated as three putative zinc fixation sites were predicted. For each condition simulated, four zinc ions were positioned based on the location of zinc binding residues such as cysteines and histidines. The predicted zinc interaction sites are magnified and the putative interacting residues side-chains are represented.

**Figure 2 ijms-25-02267-f002:**
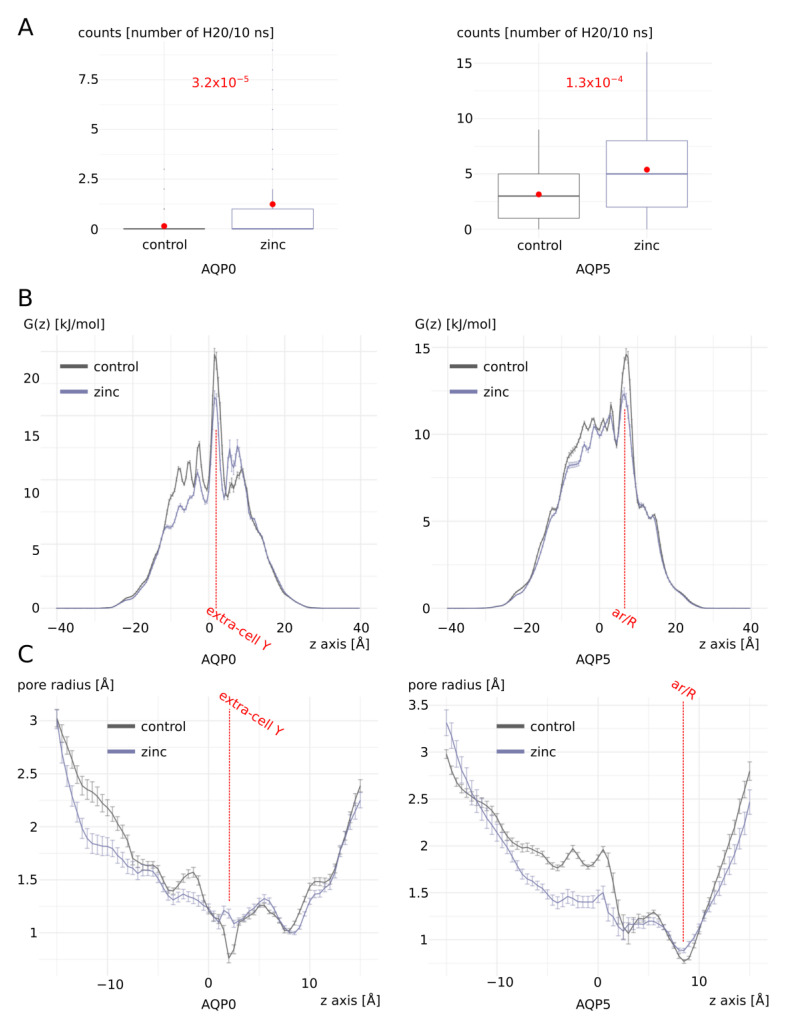
Zinc increases AQP0 and AQP5 water permeability. (**A**) Water permeability of AQP0 and AQP5 with and without zinc cations. Water permeability is estimated by the number of water molecules crossing the whole transmembrane pore section of 30 angströms of length in 10 nanoseconds. The two conditions were compared using the non-parametric Mann–Whitney test. The associated *p* values are indicated in red. (**B**) Free-energy profiles. (**C**) Pore radius. For both water counts, free-energy profiles, and pore radius profiles, 10-nanosecond sub-parts of the trajectory were used, yielding 80 statistical repetitions per condition (see methods). For each 10-nanosecond sub-trajectory, all 1000 frames were used to extract frequency of water molecule positions inside the pore for free-energy calculation, and pore radius computation was performed with HOLE software (release 2.2.005) on the 10 nanosecond-averaged AQP fold extracted with GROMACS toolbox.

**Figure 3 ijms-25-02267-f003:**
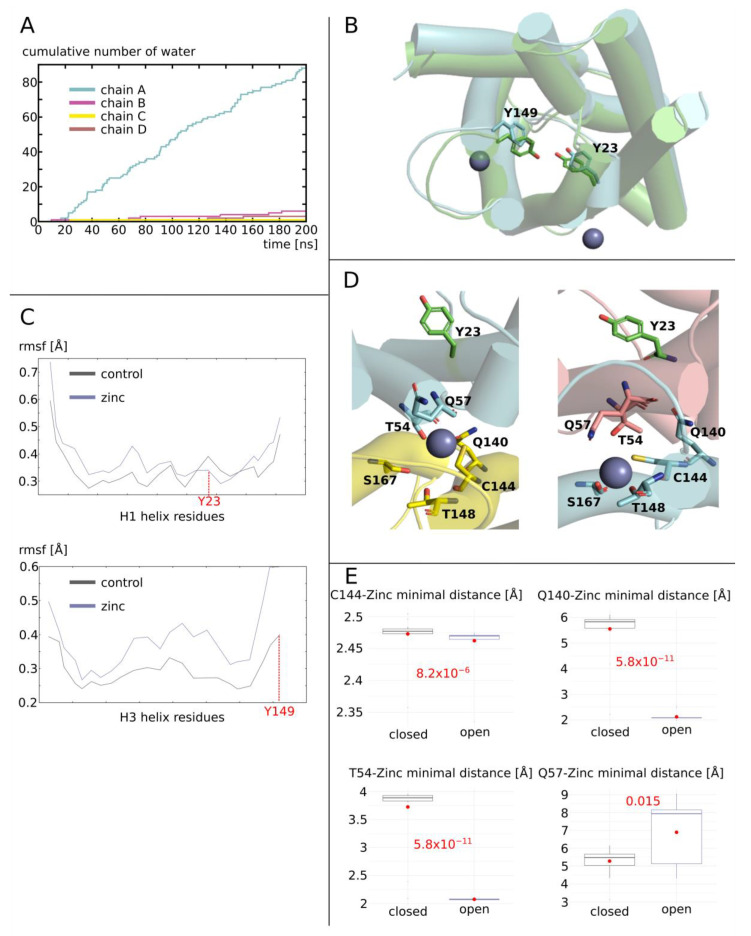
Molecular mechanism of AQP0 water permeability enhancement by zinc. (**A**) Cumulative number of water permeations (i.e., crossing the whole 30-angström-long transmembrane pore region) along the simulated trajectory for the four sub-units of AQP0 with zinc. (**B**) Schematic representation of chain A from condition AQP0 with zinc as seen from the intra-cellular compartment at the beginning (green) and at the end (blue) of the trajectory. The two conformations were aligned to one another with PyMOL. The two tyrosine side-chains responsible for AQP0 additional constrictions are represented. Zinc is pictured by spheres. (**C**) Root mean square fluctuation of helices H1 and H3 alpha carbons of chain A. Tyrosine 23 and tyrosine 149, which are responsible for the two AQP0 specific additional constrictions, are indicated in red. (**D**) Schematic representation of the interfaces between chain C and chain A and between chain A and chain D as seen from the intra-cellular compartment. On the left is depicted the interface corresponding to the open sub-unit (chain A) and on the right, the one corresponding to a closed sub-unit (chain D). Residues susceptible to interacting with zinc are represented as sticks. Chain A is depicted in blue, chain C in yellow and chain D in brown. Residue of AQP0 main constriction, tyrosine 23 is colored in green. (**E**) Minimal distance between four residues of the interface and zinc. Ten-nanosecond sub-trajectories were used for analysis yielding 20 statistical repetitions per condition. The two conditions were compared to one another using the non-parametric Mann–Whitney test. The associated *p* values are indicated in red.

**Figure 4 ijms-25-02267-f004:**
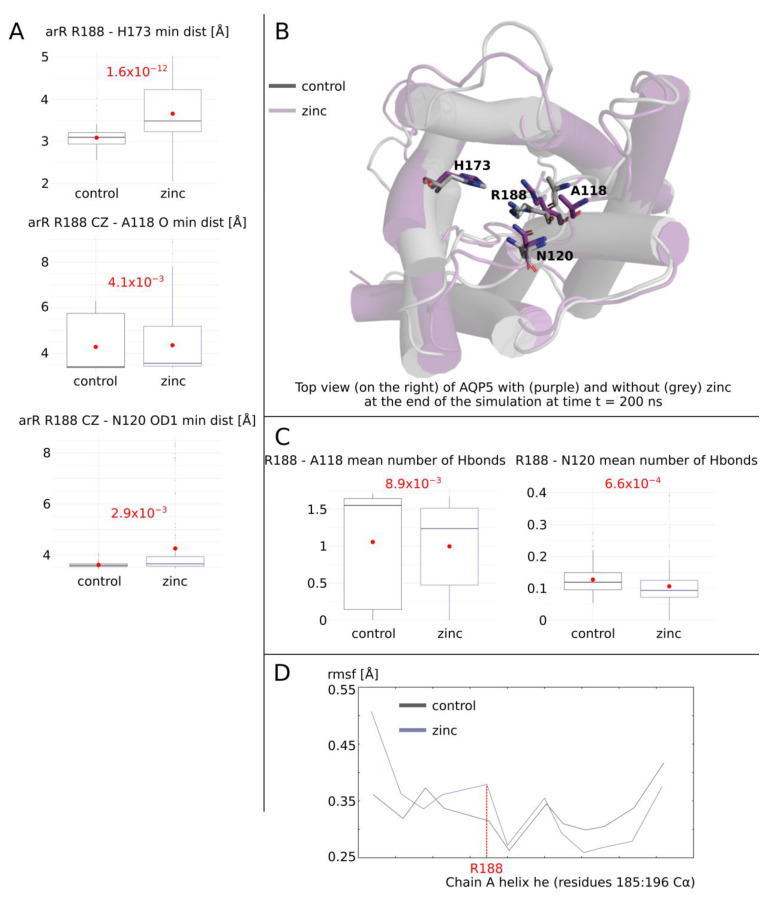
Molecular mechanism of AQP5 water permeability enhancement by zinc, part I. (**A**) Minimal distance between residues of the extra-cellular vestibule. The conditions were compared using the non-parametric Mann–Whitney test. The associated *p* values are indicated in red. (**B**) Schematic representation of the AQP5 sub-unit at the end of the simulation at t = 200 ns from conditions “control” and “zinc” structurally aligned on one another. Extra-cellular residues studied in (**A**) are represented. (**C**) Mean number of hydrogen bonds between R188 and A118 or N120 computed with the GROMACS hbond tool. The conditions were compared using the non-parametric Mann–Whitney test. The associated *p* values are indicated in red. For (**A,C**), 10-nanosecond sub-trajectories were used for analysis, yielding 80 statistical repetitions per condition. (**D**) Root mean square fluctuation of small helix HE calculated for alpha carbons only of chain A. The arginine of the ar/R constriction associated with permeability regulation R188 is indicated in red.

**Figure 5 ijms-25-02267-f005:**
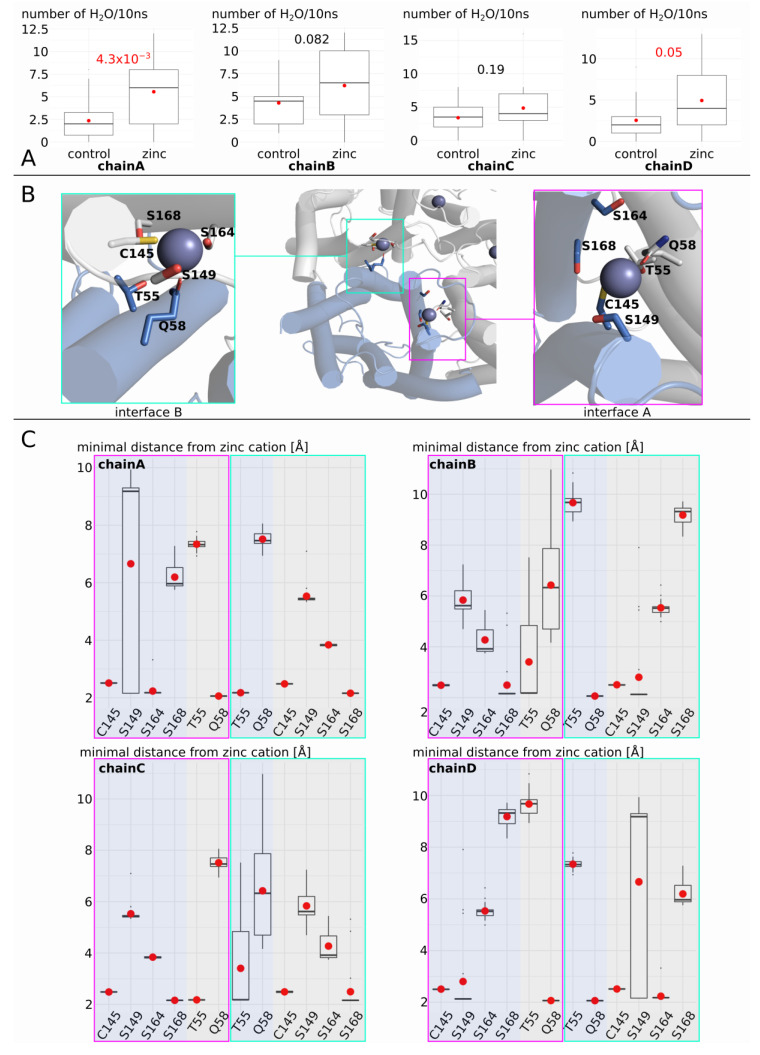
Molecular mechanism of AQP5 water permeability enhancement by zinc, part II. (**A**) Water permeability estimated through the number of water molecules crossing the whole transmembrane region (30 angströms long) in 10 nanoseconds was computed for each sub-unit, yielding 20 statistical repetitions per condition for each sub-unit. (**B**) Schematic representation of the two interfaces involving zinc cations. The residues involved in zinc binding are represented. (**C**) Minimal distance between interface residues and zinc cation. Following the same color code as in (**B**), the blue parts correspond to the considered sub-unit (chain A, chain B, chain C, or chain D). The grey parts correspond to the two adjacent sub-units. One interface is indicated by a pink box, the other by a green box. Red dots indicate the mean and black lines the median.

**Figure 6 ijms-25-02267-f006:**
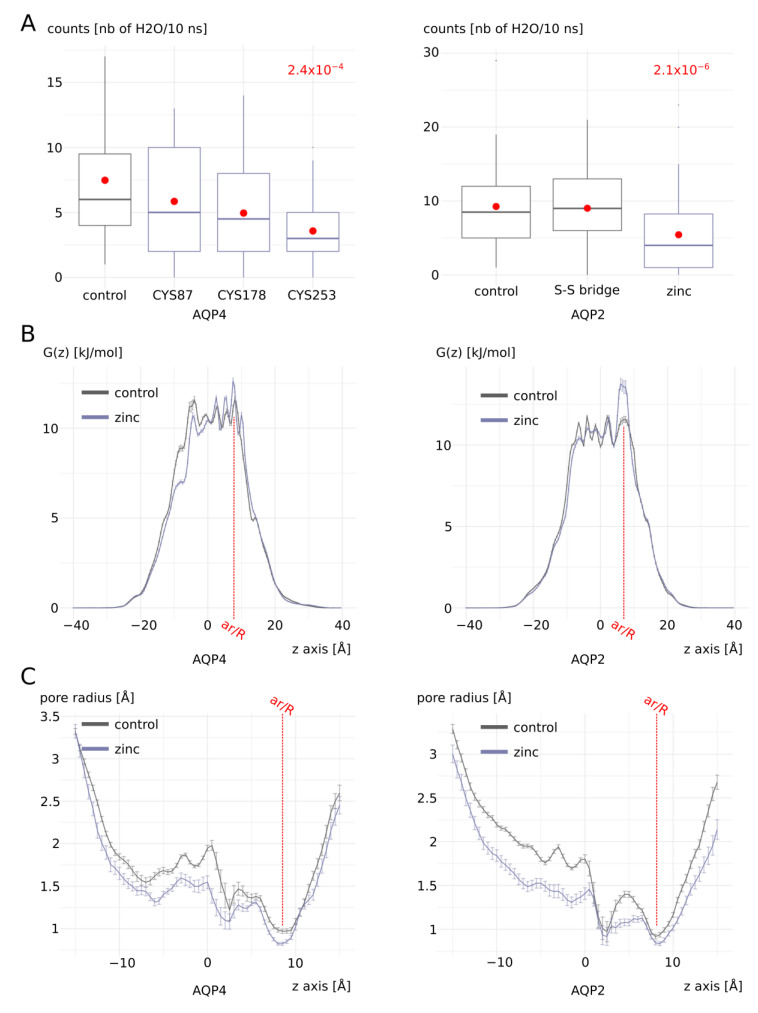
Zinc transiently decreases AQP4 and AQP2 water permeability.(**A**) Water permeability of AQP4 and AQP2 with and without zinc cations. Water permeability is estimated by the number of water molecules crossing the whole transmembrane pore section having a length of 30 angströms in 10 nanoseconds. For AQP4, three putative zinc binding sites were tested and the conditions are named after the cysteine of the site considered (i.e., CYS87, CYS178, or CYS253). For AQP2, the unique site tested included two cysteines positioned very close to each other. Thereafter, the impact of the formation of a disulfide bridge between them was also tested. The conditions were compared to one another with Bonferroni post hoc correction after the Wilcoxon test. The *p* values associated in the comparison with condition “control” are indicated in red. (**B**) Free-energy profiles. (**C**) Pore radius. For both water counts, free-energy profiles, and pore radius profiles, 10-nanosecond sub-parts of the trajectory were used, hence yielding 80 statistical repetitions per condition (see Section 4). For each 10-nanosecond sub-trajectory, all 1000 frames were used to extract the frequency of water molecules’ positions inside the pore for free-energy calculation, and pore radius computation was performed with HOLE software (release 2.2.005) on the 10-nanosecond-averaged AQP fold extracted with GROMACS toolbox.

**Figure 7 ijms-25-02267-f007:**
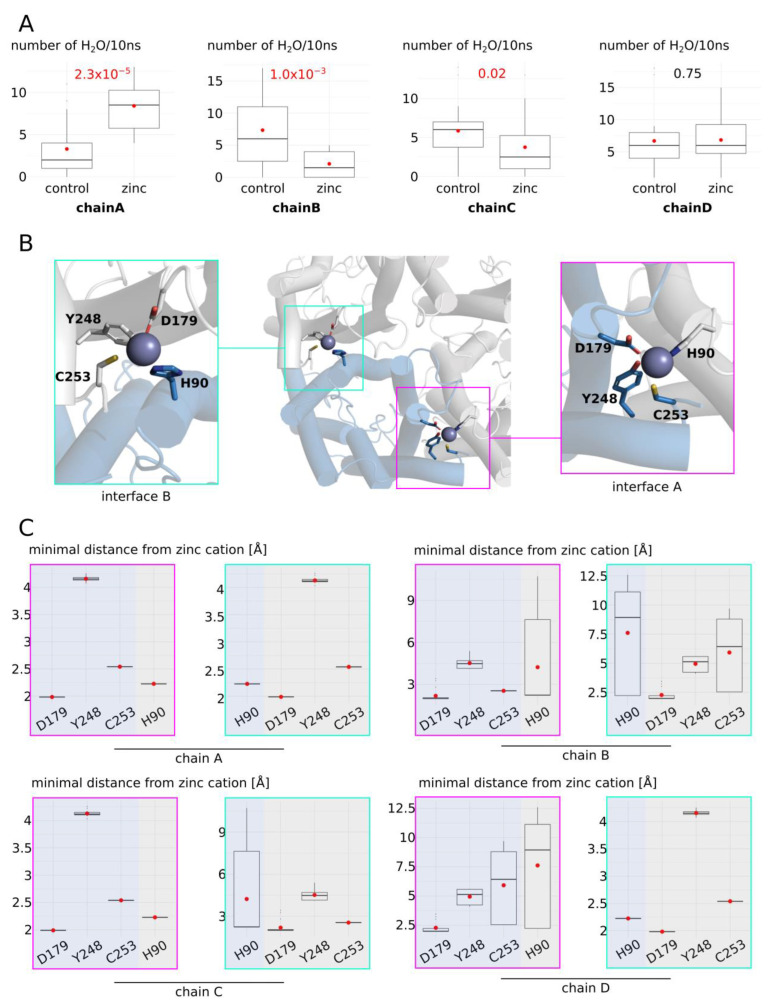
Molecular mechanism of AQP4 water permeability modulation by zinc, part I. (**A**) Water permeability estimated through the number of water molecules crossing the whole transmembrane region (30 angströms long) in 10 nanoseconds was computed for each sub-unit, yielding 20 statistical repetitions per condition for each sub-unit. (**B**) Schematic representation of the two interfaces involving zinc cations. The residues involved in zinc binding are represented. (**C**) Minimal distance between interface residues and zinc cation. Following the same color code as in (**B**), the blue parts correspond to the considered sub-unit (chain A, chain B, chain C,or chain D). The grey parts correspond to the two adjacent sub-units. One interface is indicated by a pink box, the other by a green box. Red dots indicate the mean and black lines the median.

**Figure 8 ijms-25-02267-f008:**
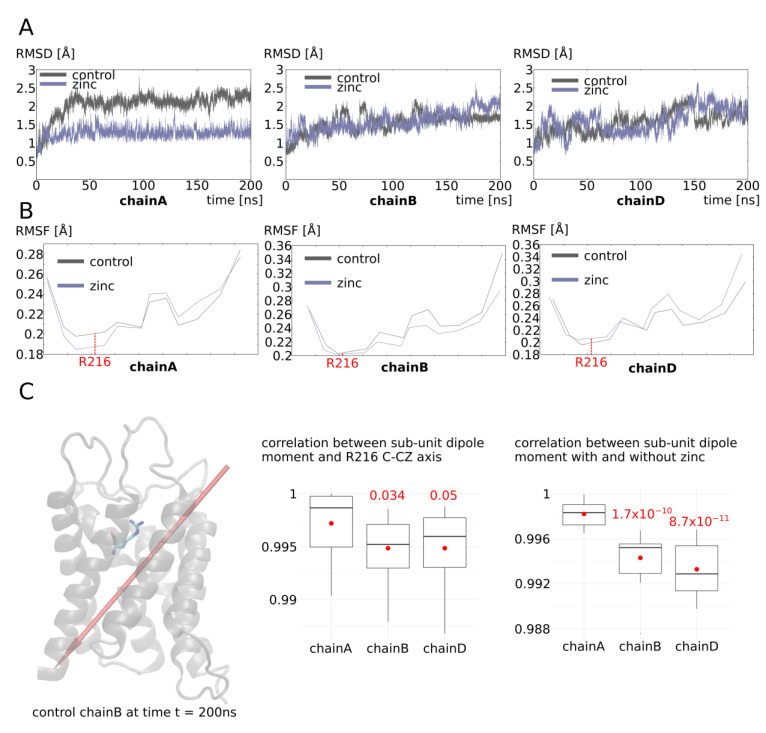
Molecular mechanism of AQP4 water permeability modulation by zinc, part II. (**A**) Root mean square deviation of chain A, chain B, and chain D backbone. (**B**) Root mean square fluctuation of small helix HE (which bears the arginine of the ar/R constriction) of chain A, chain B, and chain D. (**C**) On the left: schematic representation of chain B at the end of the simulation at time t = 200 ns. Arginine 216 of the ar/R constriction is represented. The dipole moment of the whole chain B is schematized by a red arrow. In the center: Pearson correlation coefficients between the whole chain and the associated zinc cation dipole moment and the axis formed by ar/R constriction of R216 alpha carbon and CZ carbon (within the guanidinium group). On the right: Pearson correlation coefficient between the dipole moment of the whole chain in the control condition and the dipole moment of the whole chain and the associated zinc cation in the “zinc” condition. Conditions were divided into 10-nanosecond sub-trajectories and compared using the non-parametric Wilcoxon test. Significant differences with chain A are displayed in red.

**Figure 9 ijms-25-02267-f009:**
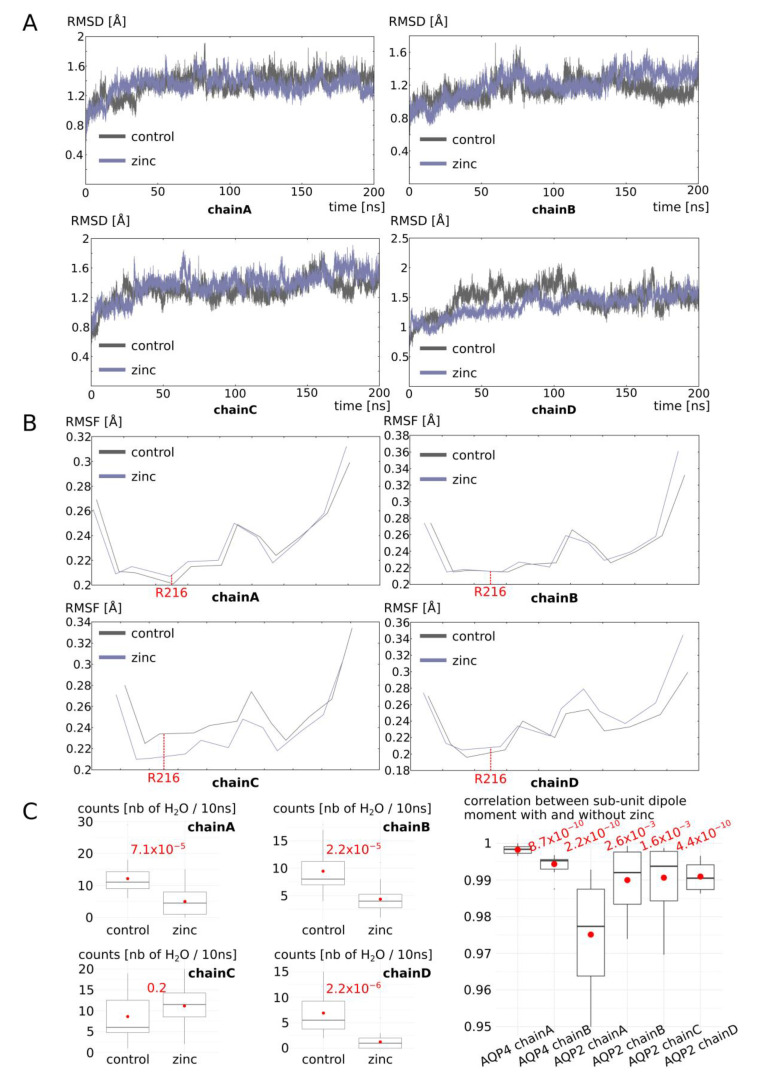
Molecular mechanism of AQP2 water permeability decrease by zinc. (**A**) Root mean square deviation of each sub-unit backbone. (**B**) Root mean square fluctuation of each sub-unit small helix HE backbone. The arginine R216 of the ar/R constriction is indicated in red. (**C**) On the left: Water permeability for each chain in the “zinc” condition compared to the corresponding chain in the “control” condition. *p*values obtained from the non-parametric Mann–Whitney test are displayed in red. On the right: Pearson correlation coefficient between dipole moment of the whole chain in the control condition and the dipole moment of the whole chain in the “zinc” condition for AQP4 chain A and chain B and for the four chains of AQP2. Conditions were compared using the non-parametric Wilcoxon test. Significant differences with AQP4 chain A are displayed in red.

## Data Availability

The data presented in this study are available on request from the corresponding author. The data are not publicly available due to the lack of a dedicated database.

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
