# Peer review of "Deciphering Molecular Mechanisms Involved in the Modulation of Human Aquaporins’ Water Permeability by Zinc Cations: A Molecular Dynamics Approach"

_ijms, 2024, doi:10.3390/ijms25042267_

Round 1

Reviewer 1 Report

Comments and Suggestions for Authors

Comments on the Quality of English Language

Author Response

Dear reviewer, on behalf of all the co-authors I would like to thank you for your constructive feedback. In the process of answering your review, we noticed a small mistake of ours concerning annotation of AQP4 sub-units: sub-unit chain C was inverted with sub-unit chain B. This has no impact on the displayed results however the figure 7, 8 and 9 as well as the associated text have been corrected.

1.1. To define putative zinc binding domains in AQP0, AQP2, AQP4 and AQP5, we chose to test all solvant accessible cysteines based on their well known involvement in zinc fingers. In order to clarify the methodology, a sentence has been added to the corresponding part in the methods and the beginning of results sections.
Indeed, without external restraints or salt bridges, zinc cations are expected to leave their putative binding site. However we observed that all zinc cations in all conditions were maintained to their putative binding site through the whole trajectories (see supplementary figure1). We consider this observation as an argument in favor of a good prediction of zinc binding by AQPs and mentioned it in the discussion section.

Concerning covalent binding with zinc, we did not mention any covalent bound formed with zinc in the manuscript. Zinc cations were manually positioned at the vicinity of cysteine residues and the atomic systems were then simulated without any external restraints applied nor covalent bounds with zinc defined.

1.2. The incorporation of a model with intra-cellular disulfide bridges formation was created hypothetically. Most disulfide bridges formation occur in the extra-cellular compartment however, intra-cellular disulfide bridges can also be found (exceptionally) in the intra-cellular environment for some proteins. Hence, because of the very close positions of the two residues to one another, we decided to test this hypothesis.

1.3. The requested information have been added to figure 2 and 6 captions.

1.4. We already aligned conformations of the beginning and the end of the 200 nanoseconds simulation of AQP0 on figure3.B. The displacement of helices is visible as well as the impact on pore residues Y23 and Y149 position. Additionally, AQP5 sub-unit from the end of the simulation extracted from conditions “control” and “zinc” were also aligned and displayed on figure 4.B. Regarding the impact on putative zinc binding site residues, figure 3.C compares functional VS nun-functional sub-units interfaces for AQP0.

2.1. You are absolutely right! Thank you for spotting this mistake of ours. We computed the mean number of Hbonds between R188 and A118 or N120 with gromacs toolbox (gmx hbond) and modified figure 4 and the associated text accordingly. A small paragraph was also added to the discussion section about the implication of loop C in AQP water permeability regulation.

2.2. We agree, we could have discuss this result further. Thereafter we added a small paragraph in the discussion section.

2.3. Indeed, we added the suggested reference in the discussion section.

2.4. Following your advice, we discussed the implications of our results for anti-metatstatic effects of zinc in the discussion.

Reviewer 2 Report

Comments and Suggestions for Authors

In this study the authors explored, by an in-silico approach, the zinc cation and AQP interactions demonstrating that the positioning of zinc cations close to specific sites was associated with a change in water permeability. Moreover, they described two different possible molecular mechanisms to account how the addition of zinc cations can lead to an increase or decrease of water permeability.

The study is quite interesting, and all things considered, the article is well written. However, due to the specificity of results and the consequent difficult to understanding for non-expert, the authors should better summarize the results at the beginning of the Discussion paragraph.

Moreover, some issues should be addressed:

·      The number of replicates should be specified for each experiment in the captions of the figures.

·      The results about AQP2 should be added to 2.3 paragraph as well as to caption of figure 6.

·      Line 390: Four has to be replaced with For

Author Response

Dear reviewer, on behalf of all the co-authors I would like to thank you for your constructive feedback.

Following your advice we synthesized our results better in the beginning of the discussion. The end part of the discussion was also amended according to the other reviewer comments.

The number of replicates has been added to each figure caption.

Results about AQP2 were added to paragraph 2.3 and to caption of figure 6. Thank you for spotting this.

Typo has been corrected.